# Knowledge and attitudes of Implementation Support Practitioners—Findings from a systematic integrative review

Leah Bührmann[1,2☯*], Pia Driessen[3☯*], Allison Metz[4], Katie Burke[1,5], Leah Bartley[4], Cecilie Varsi[1,6], Bianca Albers[1,7]

1 European Implementation Collaborative, Zurich, Switzerland, 2 Department of Social Work, Education and Community Wellbeing, Northumbria University, Newcastle, United Kingdom, 3 Aachen Institute for Rescue Management and Public Safety, Aachen, Germany, 4 School of Social Work, University of North Carolina at Chapel Hill, Chapel Hill, NC, United States of America, 5 Centre for Effective Services, Dublin, Ireland, 6 Faculty of Health and Social Sciences, University of South-Eastern Norway, Drammen, Norway, 7 Institute for Implementation Science in Healthcare, University of Zurich, Zurich, Switzerland

☯ These authors contributed equally to this work.
* leah.buhrmann@northumbria.ac.uk (LB); pdriessen@ukaachen.de (PD)

## Abstract

### Background

It requires thoughtful planning and work to successfully apply and sustain research-supported interventions like healthcare treatments, social support, or preventive programs in practice. Implementation support practitioners (ISPs) such as facilitators, technical assistance providers, knowledge brokers, coaches or consultants may be involved to actively support the implementation process. This article presents knowledge and attitudes ISPs bring to their work.

### Methods

Building on a previously developed program logic, a systematic integrative review was conducted. Literature was sourced by searching nine electronic data bases, organizational websites, and by launching a call for publications among selected experts and social media. Article screening was performed independently by two researchers, and data from included studies were extracted by members of the research team and quality-assured by the lead researcher. The quality of included RCTs was assessed based on a framework by Hodder and colleagues. Thematic Analysis was used to capture information on knowledge and attitudes of ISPs across the included studies. Euler diagrams and heatmaps were used to present the results.

### Results

Results are based on 79 included studies. ISPs reportedly displayed knowledge about the clinical practice they work with, implementation / improvement practice, the local context, supporting change processes, and facilitating evidence-based practice in general. In particular, knowledge about the intervention to be implemented and its target population, specific

**Data Availability Statement:** All corresponding materials are contained within the paper and the OSF database (https://osf.io/9kfqr/).

**Funding:** The author(s) received no specific funding for this work.

**Competing interests:** The authors have declared that no competing interests exist.

improvement / implementation methods and approaches, organizational structures and sensitivities, training, and characteristics of (good) research was described in the literature. Seven themes describing ISPs' attitudes were identified: 1) professional, 2) motivated / motivating / encouraging / empowering, 3) empathetic / respectful / sensitive, 4) collaborative / inclusive, 5) authentic, 6) creative / flexible / innovative / adaptive, and 7) frank / direct / honest. Pertaining to a professional attitude, being responsive and focused were the most prevalent indicators across included publications.

## Conclusion

The wide range and complexity of knowledge and attitudes found in the literature calls for a comprehensive and systematic approach to collaboratively develop a professional role for ISPs across disciplines. Embedding the ISP role in different health and social welfare settings will enhance implementation capacities considerably.

## Introduction

Organizational efforts supporting the implementation of research-supported interventions (RSIs) are required to bring about change effectively. Be it (health) care treatments, social support, or preventive programs—implementing RSIs often requires work at multiple levels of a system to successfully apply and sustain these complex interventions in practice [1, 2]. Implementation usually involves various parties, such as organizations, health professionals, and care recipients, and collaboration between them [3]. Thus, the implementation of RSIs is not a fast-selling item, but requires thoughtful planning, coordination and an acquisition and allocation of resources supportive to the implementation process. Such a process can be time-intensive, even more so when the goal is to implement the new RSI in a sustainable fashion [4].

Implementation support activities aim to facilitate implementation processes. While facilitation can be operationalized in numerous ways, one possibility is to provide active implementation support in the form of human resources. As opposed to either letting implementation happen or helping it happen, Greenhalgh and colleagues suggest to 'make it happen' by installing active expert support roles [5]. In the current literature, implementation support roles are often operationalized and described in the form of e.g., *facilitators*, *technical assistance providers*, *knowledge brokers*, *coaches*, or *consultants*. Albers et al. worked towards providing a more unified approach to understanding implementation support roles and conceptualize the people fulfilling those roles as *Implementation Support Practitioners* [6].

Implementation Support Practitioners (ISPs) have been defined as professionals supporting the implementation of RSIs, programs and/or policies and their sustainability in practice, by working together with stakeholders to effectively deliver such practices. ISPs can be located internally or externally to an implementing organization. They can also be part of an intermediary or purveyor organization, which are established with the aim to disseminate information about and provide assistance for the adoption of an intervention [7]. Their main goal is to build implementation capacities within the individuals and organizations they support. Implementation capacities are defined as the knowledge, skills and attitudes needed to implement innovation. Acquiring implementation capacities enables an organization to, for example, select contextually appropriate implementation frameworks, strategies or tools and apply these to different scenarios in real world health services [6].

In order to understand *how* ISPs can support the implementation of change in practice, Albers, Metz, and Burke developed a preliminary logic for ISPs [6]. This logic describes that the *implementation intervention* (the ISP role) is operating through a *mechanism of change*, namely an interplay between *resources* (position, professional background, skills, knowledge, attitudes) and *reasoning (*the evoked response in the stakeholders*)*, to generate an *output* (implementation capacities) which finally results in an *outcome* (implementation outcome, i.e., successful implementation). Whether *resources* can unfold, and *reasoning* occurs depends on the interplay between the context in which the implementation process is embedded and the active contribution of the ISP.

There have been previous efforts to describe the competencies that are required to succeed in providing active implementation support [8, 9]. However, these studies typically examine distinct ISP roles (e.g., knowledge broker or facilitators) rather than showing an overarching picture of active implementation support role competencies. A key premise of the work presented in this article has been to integrate the knowledge of competencies across comparable implementation support roles to build a shared knowledge base. Based on this premise, this work aimed to identify what (a) *knowledge* and (b) *attitudes* the literature describes as being present in ISPs and influencing their work.

## Concepts under study

Knowledge and attitudes are closely related concepts that have a central role in implementation science and practice as determinants of individual and organizational behavior. For example, attitudes are a key component of the theory of planned behavior [10], commonly used to examine implementation in health and human services [11, 12], and have shown to predict outcomes, e.g., research utilization among nurses [13]. Considerable efforts have been made to enable attitude measurement in implementation science in order to better understand its contribution to implementation processes [14–16]. Knowledge on the other hand is a central construct in determinant frameworks–such as the Consolidated Framework for Implementation Research [17]–and has been highlighted in numerous studies as a persistent barrier to evidence uptake and implementation including healthcare [18, 19], social welfare [20, 21] and education [22]. It is therefore relevant to also consider the role of these two constructs for the work of ISPs. In the following, they are described and defined in light of this role.

**Knowledge.**   In the context of implementation support, *knowledge* is defined as "*the factual information ISPs bring to and acquire about the contexts in which they work*" (p. 5) [6]. A comprehensive knowledge base—for example, knowledge about the intervention, program, or policy to be implemented; implementation concepts; or certain organizational structures—is an indispensable requisite for ISPs. Knowledge can be used as an instrument for change, and also defines individuals as influential and as experts [23]. It therefore can be seen as an active resource the ISP can draw upon and simultaneously as a passive requirement to shape stakeholders' perception of an ISP's credibility. A literature review on concepts in knowledge transfer highlights that opinion leaders need to be perceived as knowledgeable by their stakeholders and willing to share their knowledge with them [23].

Although current literature highlights the importance of a knowledge base for ISPs' work [9, 24, 25], in-depth research on a detailed conceptualization of ISPs' knowledge is yet limited. Among the few studies exploring the role of knowledge for implementation facilitation is a realist review published by McCormack and colleagues [26]. It suggests that the combination of academic and local knowledge and practical experience in change agents is important for successful knowledge translation [26]. Moreover, the above-mentioned literature review describes that knowledge of opinion leaders is often product- and situation-specific [23].

**Attitudes.**  According to social psychology, *attitudes* are "*a relatively enduring organization of beliefs, feelings, and behavioral tendencies towards socially significant objects, groups, events or symbols*" (p. 150) [27]. These tendencies are expressed "*by evaluating a particular entity with some degree of favor or disfavor*" (p. 1) [28]. Hence, ISPs' attitudes can be defined as the cognitive predispositions that influence their implementation support-related activities and decisions [6, 29]. In the literature, multiple attitudes have been assigned to facilitators, including the willingness to think innovatively and positively and be curious and open to suggestions [9, 30, 31]. A generally positive attitude of change agents affecting knowledge utilization is highlighted in a realist review of interventions to promote evidence-informed healthcare [26]. In a scoping review of facilitation roles (p. 6) [25], champions were described as "*driven*" [32] and "*passionate about their work*" [33], and characterized by an overall belief in the necessity of the change to be implemented [34, 35]. Similarly, Metz et al. [29] describe commitment as a core principle among ISPs, which may include patience, resilience, and the willingness to challenge the current status quo.

**Aim.**  This research is part of an ongoing international collaborative program of work on implementation support practice and practitioners involving the National Implementation Research Network (NIRN), University of North Carolina, the European Implementation Collaborative (EIC) and the Centre for Effective Services (Ireland). This collaboration involved the conduct of a systematic integrative review of studies reporting on multiple aspects of the work of ISPs, including ISP *skills* [36] and the *mechanisms of change* that may explain how ISP efforts generate change in others [37]. The present article reports on ISP *knowledge* and *attitudes*.

The aim of this third and final part of the review was to describe both the knowledge base and the mindset ISPs utilize in their work, thereby complementing existing work in this field based on an accumulated definition of the role of an ISP (as opposed to separate roles) and addressing the distinct features of ISP *knowledge* and *attitudes* (as opposed to *characteristics* in general). Together, all findings of this review will help to refine the previously developed ISP program logic [6] and contribute to theory building in the field of implementation science and practice.

## Methods

A systematic integrative review method was applied to consolidate findings on ISP knowledge and attitudes from a diverse body of literature. The overarching review was conducted in five stages [38]: 1) problem identification, 2) literature search, 3) data evaluation, 4) data analysis, and 5) data presentation. These five stages are outlined below, with the focus on the study of *knowledge and attitudes* of ISPs. A detailed description of the overall methods used to conduct the review is provided in Albers et al. [36] and in the respective electronic results addendum (S1 Appendix, pp. 1–2) [39]. The review was conducted by a team of five researchers with expertise in systematic literature reviews (BA and CV), content expertise related to social work (BA and LBa), health services (CV, PD, and LB), and education (BA).

Our methodology and findings are reported in accordance with the Preferred Reporting Items for Systematic Reviews and Meta-Analysis (PRISMA and PRISMA-S) guidelines and checklists (S2 Appendix) [40]. As the corresponding review materials have already been made public via Open Science Framework [39] it is not registered elsewhere.

### Problem identification

Discussions in the research team based on a preliminary ISP program logic developed by Albers et al. [6] resulted in the agreement that the current understanding of which *knowledge*

is required, and which *attitudes* are beneficial for the work of ISPs is limited (question under study).

## Literature search

To achieve a representative collection of literature about the question under study, five search strategies were used to identify relevant publications: 1) systematic search of nine electronic data bases (ASSIA, CINAHL, Criminal Justice Abstracts, ERIC, Family and Society Studies Worldwide, Medline, PsycInfo, Scopus and SocIndex), 2) search of organizational websites, 3) call for publications among selected experts, 4) open call for publications through social media (i.e. Twitter, LinkedIn), and 5) reference check for all included studies. Searches were conducted between February and April 2019. The search string for the systematic search of the electronic data bases as well as the detailed inclusion and exclusion criteria applied are described in S1 Appendix (pp. 1–4) and in Table 1, respectively.

**Table 1. Overview of inclusion and exclusion criteria.**

| | Included | Excluded |
|---|---|---|
| **Study designs** | Any primary study design, including, e.g.:<br>• Randomized, controlled studies<br>• Quasi-experimental studies<br>• Pre-post evaluations<br>• Case studies | • Systematic reviews of any study designs<br>• Theoretical / conceptual studies<br>• Expert commentary<br>• Opinion pieces<br>• Conference proceedings<br>• Books / book chapters<br>• Editorials<br>• Other publications that are / report "non-studies"<br>• Dissertations / theses |
| **Population** | • The target population(s) of the study are implementation specialists. | • The target population(s) of the study are current or future *implementation researchers* or other non-ISPs. |
| **Sectors** | • Health care (including mental health)<br>• Social care / social welfare (including child welfare, aged care, labor market services, crime and justice, family services etc.)<br>• Education (including ECE, primary, secondary and high school services) | • IT<br>• Environmental services<br>• Any other service sector not listed in the left column. |
| **Content** | • The study focuses on supporting the implementation of a research-supported program, practice or policy AND<br>• the study involves a person or a team or an intermediary organization providing implementation support to others AND<br>• the study provides information about / conceptualizes / assesses implementation know how / skills / capacity<br>• Examples of interventions that could be of relevance:<br>• Implementation training<br>• Implementation coaching<br>• Technical assistance<br>• Implementation coaching<br>• Implementation facilitation<br>• Implementation policies<br>• Implementation mentorship | • Studies focused on the specific knowledge, skills, capacities required to implement e.g. specific *clinical interventions*<br>• Studies focused on factors that support successful practice-science / community-university partnerships<br>• Studies focused on online solutions only (repositories providing electronic implementation resources and nothing else) |
| **Geography** | • High income countries<br>• Low / middle income countries | No exclusion based on geography |
| **Languages** | • English<br>• German<br>• Danish<br>• Swedish<br>• Norwegian | Any other language not listed |
| **Publication date** | No limitations | No limitations |

## Data evaluation

The researchers double-screened the titles, abstracts, and full-texts independently using the platform *Covidence*. Conflicts were solved by a third member of the team. Data was extracted from the included studies by using a standardized data extraction form consisting of 19 items (i.e., study design and aim, method, geography, sector, setting, sampling strategy, sample size, clinical intervention, ISP information, outputs, and outcomes). Data from each included study was extracted by one research team member. The quality of the data extraction was assured by the lead author.

The quality assessment of included studies was focused on randomized controlled trials only. This decision was taken by the research team considering the purpose and character of this review. Based on a framework by Hodder et al. [41] all included RCTs were quality assessed regarding consistency in terminology use, integration of theory, and consideration of benefits and harms of implementation interventions. The quality assessment tool as well as the results are included in the S1 Appendix (pp. 20–22).

## Data analysis

For the analysis of key information about ISPs, included studies were uploaded to the online qualitative data analysis platform *dedoose* [42]. While the data analysis included a wide range of information about ISPs, this article reports on the data focused on *knowledge and attitudes* only. Thematic Analysis [43–45] was used to capture patterns and themes across all textual data describing *knowledge* and *attitudes* of ISPs. Thematic Analysis followed six steps: (1) data familiarization; (2) systematic data coding; (3) theme development; (4) review; (5) consolidation; and (6) reporting. Each included study was reviewed as part of the systematic coding, a process that occurred in two rounds. After an initial familiarization with the data (1), open, inductive coding (2) was used to identify all textual information describing particular knowledge or attitudes that ISPs use in their work. The focus was on the semantic level of texts, i.e., the direct linguistic meaning of words. Brief instructions, outlining what to pay attention to when reviewing texts and providing examples of different types of knowledge and attitudes, functioned as a guide for this step. Text excerpts for both categories were exported from *dedoose*, resulting in two separate spreadsheets, including all information on ISP knowledge and on ISP attitudes. The spreadsheets were reviewed separately to quality assure all coding and highlight key content for each excerpt. The second round of coding–data comparison– involved further inductive coding at the conceptual level of text information moving beyond the purely linguistic meaning of words and instead identifying broader themes (3) present in the literature. All text excerpts were reviewed in detail (4), preliminary ideas for themes were identified and refined through constant comparison with other text excerpts belonging to the same category and theme. This process resulted in distinct categories of types of *knowledge* and *attitudes* of ISPs and their detailed operationalization (5). The results were synthesized narratively and are reported in the following (6).

## Data presentation and exploration

The results of the data analysis are summarized in the present article including several visualizations (i.e., Euler diagrams and heatmaps). Euler diagrams are a graphical depiction of relationships between sets or groups, where circles represent distinct sets and their overlap illustrates a relationship between those sets [46]. The diagrams were developed based on the total amount of studies reporting a distinct category of knowledge or attitudes (size of circles) and the total amount of studies reporting on two or more distinct categories of knowledge or attitudes (overlap between circles). Heatmaps were developed by comparing the number of

included studies reporting on the position (i.e., internal, external, internal-external, research) of ISPs with the number of included studies reporting on certain types of knowledge and attitudes.

## Results

The integrative review identified 79 studies which reported on knowledge and/or attitudes of ISPs [47–125]. Fig 1 shows the flow of studies through the review process. S3 Appendix provides an overview of the main study characteristics of the included studies as well as an indication of the degree to which the included studies reported on information about ISP's knowledge or attitudes or both.

### What background do ISPs have?

School teachers, social workers, psychologists, psychiatrists, nurses, physicians, dietitians, occupational therapists, physiotherapists, epidemiologists, and pharmacists were the most dominant education descriptors across ISPs. Less specific markers used to describe this specialist status were "clinically trained" [69], "clinical practitioners" [94] or with "clinical

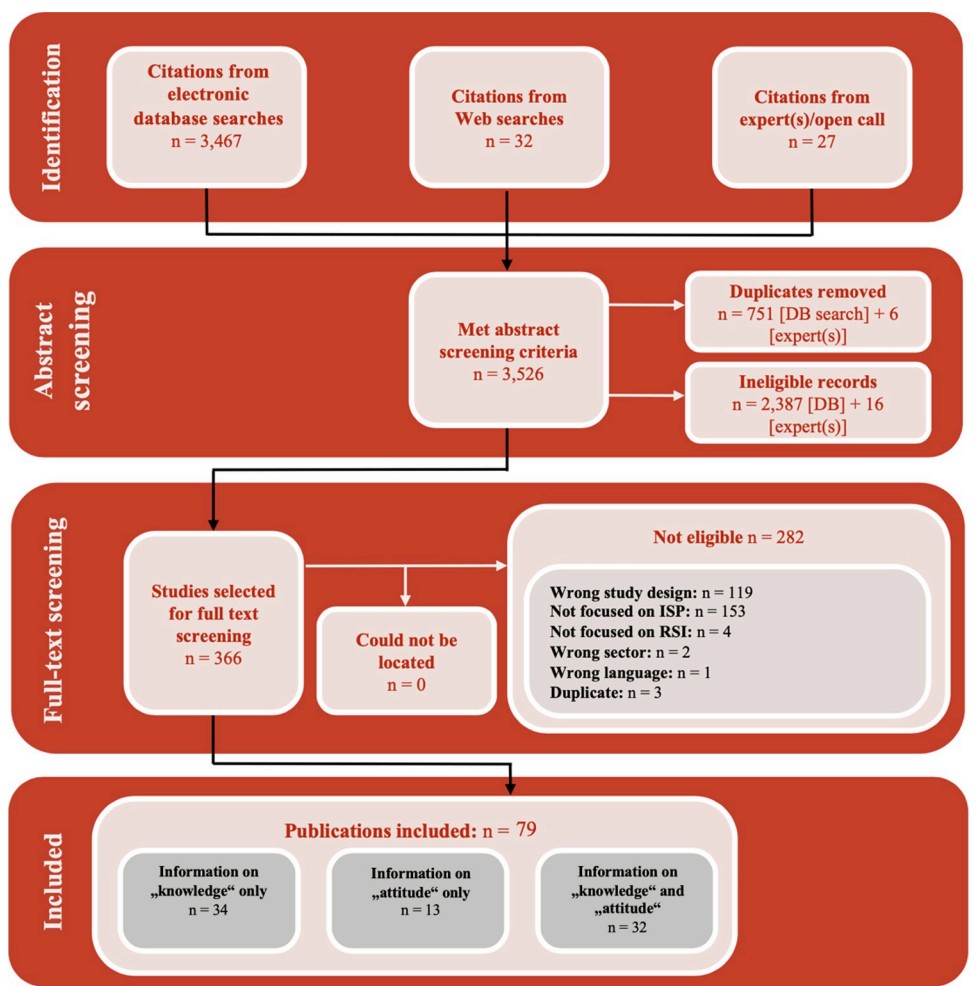

**Fig 1. Flow of studies through the review process.**

expertise" [120]. Generalists with an educational background in, for example, anthropology, political science, or organizational change were rarely used as ISPs. Only seven studies described e.g., "non-clinicians with an interest in knowledge transfer" (p. 2) [120] or individuals with project or data management background [100] as involved in implementation support, and in these cases, they were part of a set up that also included clinical specialists.

ISPs were generally described as experienced as they had worked in their profession for some time. While few studies detailed this experience in years, those that did, portrayed ISPs as having a minimum of three years' experience of working in the field in which they provided implementation support [63, 101, 119], oftentimes exceeding even 10 years of experience [77, 99, 101, 116, 119].

## What knowledge do ISPs display?

Sixty-six papers reported on ISP knowledge use. Across these publications, five types of knowledge were identified, including ISPs' knowledge about *1) the clinical practice, 2) implementation / improvement practice, 3) the local context, 4) supporting change processes, and 5) facilitating evidence-based practice (EBP) in general.* Fig 2 visualizes these findings. Each of the five themes consists of several sub-themes which are reported in the following to convey rich detail on the types of knowledge that ISPs rely on as part of their work. A comprehensive

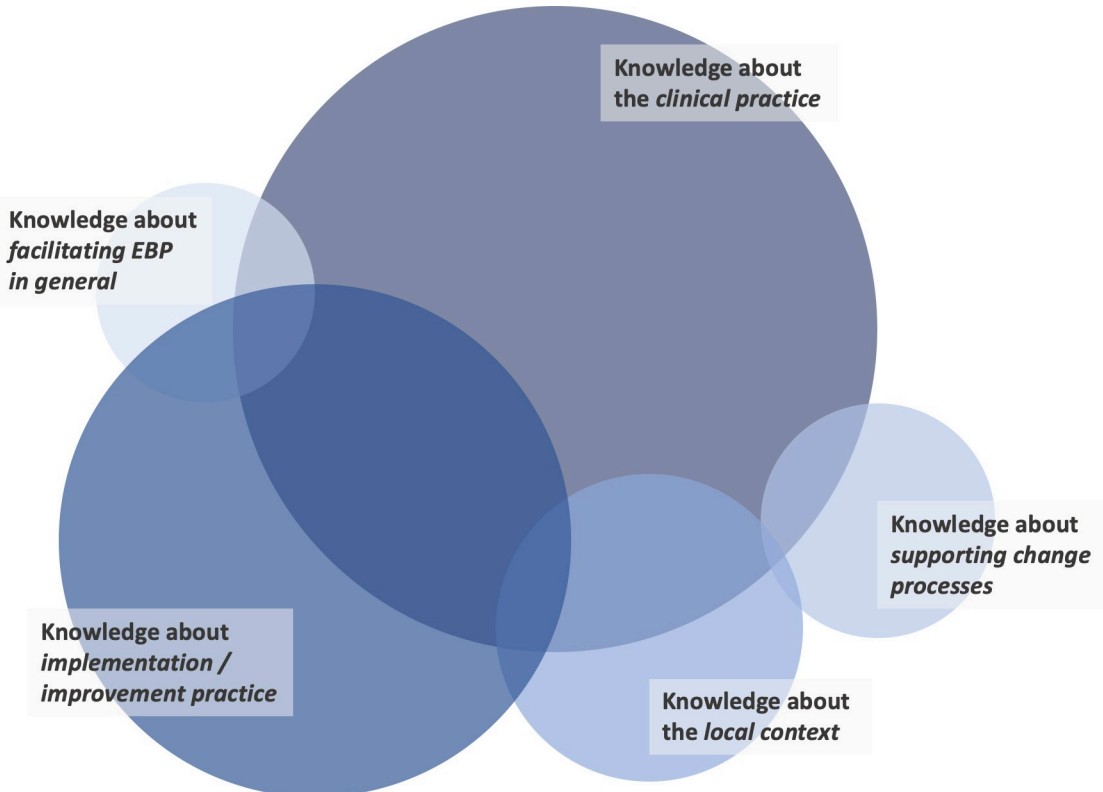

**Fig 2. The five most present types of knowledge of ISPs in the included articles.** The size of the circle indicates the number of studies which reported the specific knowledge area: knowledge about the clinical practice (n = 49), knowledge about implementation / improvement practice (n = 39), knowledge about the local context (n = 24), knowledge about supporting change processes (n = 20), knowledge about facilitating evidence-based practice (EBP) in general (n = 16). Publications were included in this visualization when they reported on at least one type of knowledge. The overlap between circles shows the co-presence of knowledge areas.

overview of the types of knowledge that were described in included publications, together with a reference to the article in which these descriptions appeared, is provided in S4 Appendix.

**Knowledge about clinical practice.** Of 49 publications providing information on knowledge about clinical practice, 36 highlighted that ISPs use knowledge about the intervention to be implemented and its target population. Knowledge about the broader research in the clinical area in which ISPs operated was described in 20 publications, e.g., by presenting ISPs as providing "information such as articles, standardized assessments of participation and other evidence-based materials" (p. 443) and facilitating the integration of this research evidence into learning processes [48].

**Knowledge about implementation / improvement practices.** Information about ISPs' use of implementation and/or improvement practice knowledge could be extracted from 39 publications. Fourteen of these suggest a need for ISPs to be familiar with specific implementation/improvement approaches, such as walk-through exercises, flow charting, nominal group technique, Plan-Do-Study-Act cycles, and process mapping. Further types of knowledge referred to in four publications each were knowledge about the principles of high-quality implementation and facilitation and about one's own role as ISP. The latter was especially notable in studies of knowledge brokers, which emphasized the necessity of ISPs understanding "the purpose of their activities in promoting knowledge and use of the tools" (p. 1583) [116], "the primary functions of their role" (p. 61) [107], or their "roles and responsibilities" (p. 5) [77]. This "understanding of own role" information was not included in studies of other ISP roles.

**Knowledge about the local context.** Being knowledgeable of the local context for an implementation effort emerged as another type of knowledge described as central to the work of ISPs across 24 publications. Most prominent was knowledge about organizational structures and sensitivities discussed in nine publications, as well as about local conditions mentioned in six publications. Both refer to knowing the facilities, staff and the physical environment in which implementation support is provided, and to being familiar with existing hierarchies, work cultures, workflows, policies and values [109, 115, 121, 125]. The latter is also described as "organizational structure and climate" by Saldana and Chamberlain [78].

**Knowledge about supporting change processes.** Further 20 studies described ISPs as using knowledge about change processes and supporting these. Five described ISPs as being knowledgeable about how to train and educate others. Training provided by ISPs focused on e.g., "the importance of program fidelity; group facilitation skills, marketing and recruiting techniques; and the participant consent and evaluation data collection processes" (p. 1009) [75]. Knowing how to effectively communicate and how to lead or facilitate groups including participation in group processes, decision-making, and conflict resolution was mentioned by three publications respectively.

**Knowledge about facilitating EBP in general.** Information on facilitating EBP in general were provided by 16 publications. EBP is defined as the process of "integrating individual practice expertise with the best available external evidence from systematic research as well as considering the values and expectations of clients" (p. 289) [126]. In this context, RSIs are considered the best available external evidence. While five articles presented knowledge about characteristics of quality research, knowledge about the key principles of EBP, knowledge translation, together with general principles of high-quality practice and strategies for sourcing evidence were mentioned by three publications respectively. For instance, Aasekjær et al. [94] describe a post-professional program that taught the different steps in EBP, based on which ISPs reported to have gained "insight into the use of research in healthcare decision making" (p. 6) and an understanding of how "to acquire practical means by which they could further improve clinical practice" (p. 6).

## Does the knowledge of ISPs differ according to their position?

ISPs can occupy different positions when providing implementation support. They may be part of the implementing organization and/or team (internal), or they may be external providers of implementation support, belonging to a different organization than those supported. In some studies, implementation support was provided through groups of ISPs, some of which were located internally while others were externally recruited. Furthermore, multiple studies included ISPs belonging to a research organization providing implementation support as part of a research study. Different locations might influence the competencies the ISP needs to work effectively. Fig 3 visualizes the relationship between types of knowledge and positions of the ISP in terms of their co-occurrence in the included studies. A darker blue field indicates that a certain type of knowledge was predominantly reported in publications reporting on a certain position of the ISP in the implementation process. Knowledge about the clinical practice and about implementation/improvement practices was reported across studies describing all different ISPs positions (internal, external, internal-external to the organization, or in research), with a slight emphasis on clinical practice knowledge for ISPs located in research and implementation/improvement practice knowledge for ISPs positioned internally-externally. Knowledge about facilitating EBP was most prominent in publications focused on ISPs with an internal position. Studies on externally positioned ISPs and those connected to research teams reported knowledge about clinical practice, implementation/improvement practice and the local context (first three domains in Fig 3) more often than knowledge about the facilitation of EBP and change processes (last two domains in Fig 3).

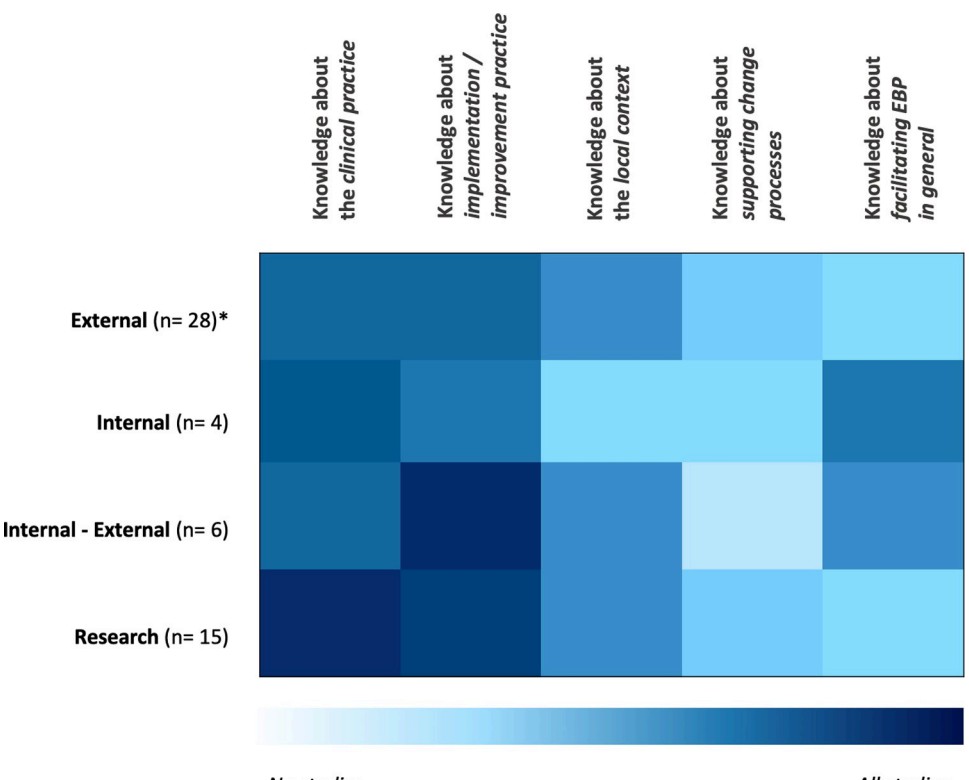

**Fig 3. The five most present types of knowledge of ISPs in the included articles according to distinct position.**
*n = number of reported studies; positions were included with a minimum of n = 4 reported studies.

## What attitudes do ISPs display?

One hundred and forty-six text excerpts about ISP attitudes described as influential to their work could be extracted from 45 publications. Across these, seven broad categories of attitudes emerged: *1) professional*, *2) motivated / motivating / encouraging / empowering*, *3) empathetic / respectful / sensitive*, *4) collaborative / inclusive*, *5) authentic*, *6) creative / flexible / innovative / adaptive*, *7) frank / direct / honest*. Fig 4 visualizes those seven most present types of ISP attitudes. A comprehensive overview of the types of attitudes that were described in included publications, together with a reference to the article in which these descriptions appeared, is provided in S5 Appendix.

**Professional.**   Twenty-eight publications described a *professional* attitude as central to ISPs' work. Detailed descriptors of this professionality were an ability to stay focused, on e.g., the key focus population, stakeholders, or intentions behind an implementation initiative [122, 125], or to help maintain its momentum [113, 121]. It also implied for ISPs to be consistent, e.g., in their approach to promoting an intervention [103], and to be responsive, e.g., to the needs and sensitivities of stakeholders [82, 95, 99], including being accessible when called for [86, 91, 108]. Few studies discussed the importance of putting boundaries around this responsivity, ensuring, e.g., that ISPs only cover needs for which no local capacities exist [118]. Part of professionality was also to be perceived as credible [93], e.g., on the basis of clinical expertise [106], an understanding of the RSI in focus [91] or of the local context into which

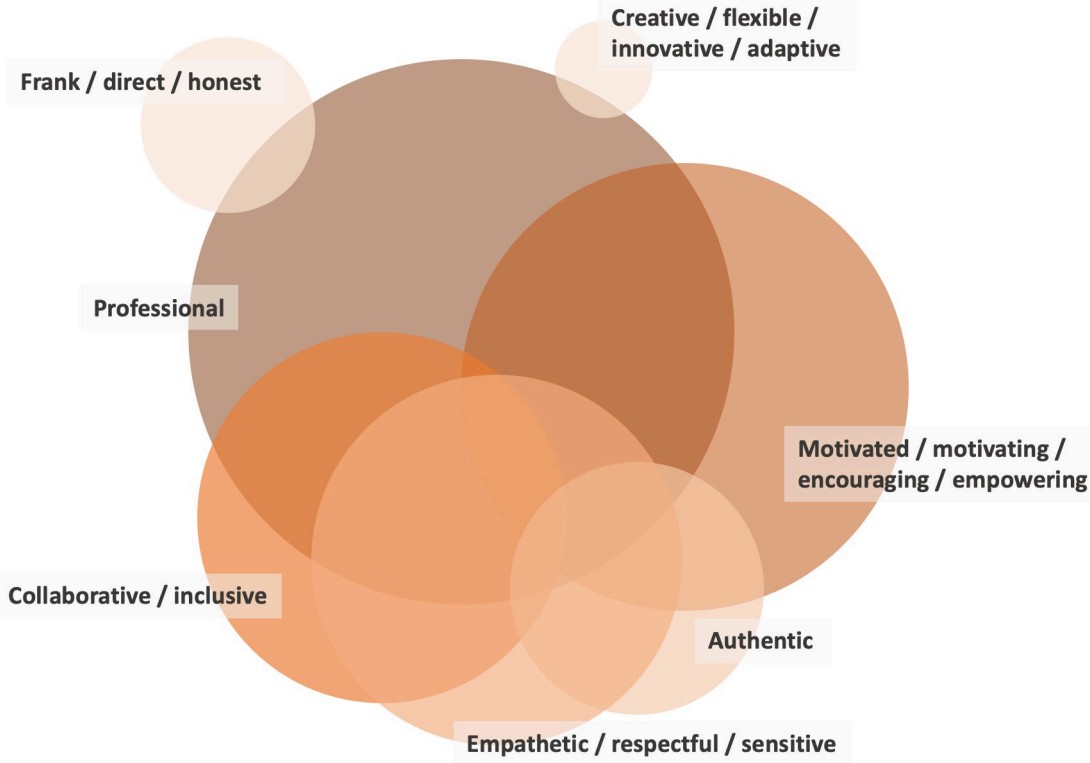

**Fig 4. The seven most present attitudes of ISPs in the included articles.** The size of the circles indicates the number of studies which reported the specific attitude: professional attitude (n = 28), motivated / motivating / encouraging / empowering attitude (n = 25), empathetic / respectful / sensitive attitude (n = 20), collaborative / inclusive attitude (n = 19), authentic attitude (n = 13), creative / flexible / innovative / adaptive attitude (n = 9), frank / direct / honest attitude (n = 5). Publications were included in this visualization when they reported on at least one type of attitudes. The overlap between circles shows the co-presences of attitudes as reported in the articles.

this RSI was implemented [119]. Studies of facilitators, knowledge brokers and intermediaries also highlighted resilience and persistence as necessary features of a professional attitude, described as a need to be "thick skinned" [101], standing up to "peers, over whom one does not have official power" (p. 1020) [113], and enduring a lack of support [121] or weeks of no visible progress or success [84].

*Motivated / motivating / encouraging / empowering.* A *motivated / motivating / encouraging / empowering* attitude of ISPs was described in 50 text excerpts from 25 studies. A consistent element described for this mindset was enthusiasm–for research in general [107], the RSI that ISPs supported in particular [99, 111], the broader project in which they were involved [100, 101, 113], and for their own role as ISP [106]. It was often this enthusiasm that was presented as a fuel for the encouraging mindset, one that could be shared with those supported [106] or even compensate for a lack of enthusiasm elsewhere in the organization [121]. Reflective of an encouraging ISP attitude was being appreciative of strengths and successes present in the implementation context [97, 100, 102, 110], providing constructive, motivating feedback and support [83, 87, 94, 106], validating and encouraging the work of implementers [108, 119, 123] and providing "the energy and direction to stay on track" (p. 422) [89]. As such, the encouraging attitude focuses on the positive.

**Empathetic/Respectful/Sensitive.** In 20 studies, ISPs were described as pursuing an *empathetic / respectful / sensitive* attitude, mirrored in a sensitivity toward the challenges and barriers faced by the stakeholders they assist. This implied recognizing the uniqueness of different implementation contexts [85], listening actively [90], being non-judgmental [119] and non-authoritative [92, 112] in their work, sometimes also providing emotional support [119], and meeting stakeholders at whatever developmental stage they were in regarding their change process [118].

**Collaborative/Inclusive.** A *collaborative / inclusive* attitude was reported as a central ISP characteristic in 19 studies. It represents a willingness to recognize and actively engage a broad diversity of individual or organizational stakeholders and their perspectives in the implementation support process. In studies, this willingness was described as "an interest in others" (p. 13) [85] displayed in inclusive efforts to involve stakeholders in the implementation and to listen to their perspectives with an open mind [94, 100, 101, 118]. It also showed in actively engaging with stakeholders [106], and in continuously building new relationships [117], thereby adding to the relational resource ISPs represented for their stakeholders [111, 125]. In the support of those most directly involved in an RSI implementation, pursuing a collaborative mindset could also show in ISP efforts to tone down one's role as an expert and instead position oneself as a "partner" to those supported [119], for example, by continuously inviting stakeholders to provide input [100], to define the agenda for the support required [95], and to "share thoughts before the coach shares" (p. 33) [92]. Few studies discussed the potential tensions that might emerge from this collaborative attitude, for example for ISPs who also hold an evaluative role and monitor fidelity and other implementation performance data for those they support [95, 108].

**Authentic.** Further 13 studies described an *authentic* attitude to be important for ISPs, representing a state in which ISPs are prone to be real, true, or genuine in their work. Authenticity was described in primarily two ways, both of which were presented as also contributing to ISPs' credibility among their stakeholders: 1) ISPs went into the field and supported stakeholders directly at the frontline of service delivery [96, 105, 119] or 2) ISPs embodied the knowledge and methods they taught others and used these directly in their implementation support techniques [85, 95, 108, 109, 113]. The former was described as a willingness and ability to support implementation directly in, e.g., classrooms or clinics through "hands on" support, demonstrating how an RSI can be applied in real practice. The latter was reported as

ISPs' willingness and ability to "walk the talk" by role-modelling the knowledge and techniques that they wanted others to use in their own coaching, consulting, or facilitation, for example, the principles of EBP or Motivational Interviewing. Other aspects of authenticity covered in studies were ISP self-awareness [121], willingness to give the implementation support provided "a personal touch" (p. 110) [115] and to genuinely empathize with stakeholders based on a true understanding of the realities of practice [123].

**Creative/Flexible/Innovative/Adaptive.** Nine studies included descriptions of a *creative / flexible / innovative / adaptive* attitude as a key characteristic of ISPs, referring to the ISP being flexible in their approach to supporting implementation processes and tailoring this approach to different stakeholders and contexts. This includes to accommodate the preferences and constraints that individuals, organizations and systems present as part of an implementation initiative [115]. In displaying flexibility, ISPs ensure making information and support relevant and useful to their stakeholders; they source, use and combine implementation resources as well as strategies in creative ways [88, 98], "think outside the box" (p. 135) [101], and remain curious [111, 116] throughout an implementation process. This curiosity was described as being supportive of their own learning, about new ISP techniques and tools, thereby further expanding their flexibility. None of the included studies discussed how ISPs manage potential tensions existing between a flexible attitude and the need to ensure that the implementation of RSIs adheres to specific quality criteria.

**Frank/Direct/Honest.** Five studies indicated ISPs to have a *frank / direct / honest* attitude, which refers to a certain straightforwardness when working with stakeholders involved in the implementation of RSIs. For instance, Akin et al. [81] presented authenticity and transparency as two important interrelated attitudes. Olson et al. [92] use the term "forthright" (p. 33) to describe a similar attitude, suggesting that ISPs should be direct when working with their stakeholders. Similarly, Waterman and colleagues highlight that "honesty, directness and focusing on what needed to be done were effective in helping staff to persist" (p. 9) [122].

## Do the attitudes of ISPs differ according to their position?

Fig 5 shows that studies reporting on ISPs located internally-externally to the organization also report predominantly on an *empathetic / respectful / sensitive* attitude. A *professional* attitude is mostly reported in studies of ISPs who are located externally or internally-externally. No other attitude is particularly highlighted in relation to a specific position of the ISP.

## Discussion

Based on a systematic, integrative review, distinct types of knowledge and attitudes were identified that ISPs reportedly use in their work. This includes five types of knowledge: *knowledge about the clinical practice, implementation / improvement practice, the local context, supporting change processes, and facilitating EBP in general*. Particularly salient was ISPs' use of knowledge about the RSI to be supported in its implementation and its target population, specific improvement / implementation methods and approaches, organizational structures and sensitivities, training, and characteristics of (good) research. Seven distinct attitudes were identified that characterize ISPs: *motivated / motivating / encouraging / empowering, empathetic / respectful / sensitive, collaborative / inclusive, authentic, creative / flexible / innovative / adaptive*, and *frank / direct / honest attitude*. A *professional attitude* was the most reported, with being responsive and being focused as the most prevalent indicators across included publications. The order of mentioned attitudes represents the frequency with which they were reported in the literature. Attitudes were often identified in combination, meaning that single publications reported on several attitudes of ISPs at a time.

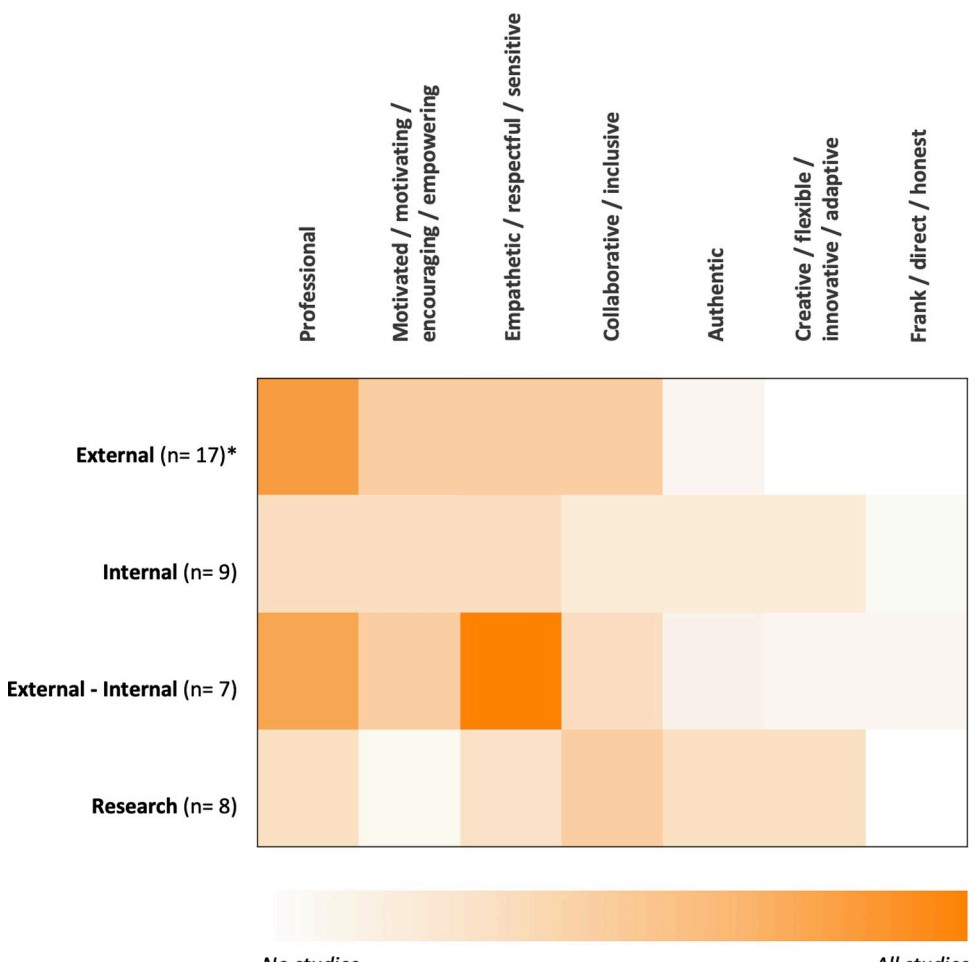

**Fig 5. The seven most present attitudes of ISPs in literature according to distinct position.** *n = number of reported studies; positions were included with a minimum of n = 4 reported studies.

These findings are aligned with those from a previous review [25] describing that a broad range of facilitating roles depends on being highly *knowledgeable* in relation to their clinical area, context and the research relevant to their work. Similarly, understanding context, research processes, knowledge translation and dissemination, and EBP were highlighted as needed by ISPs in a scoping review of the literature on knowledge translation [9]. These two previous reviews also describe what this review understands as attitudes. Mallidou et al. [9] identified five key attitudes of knowledge translators including *being confident, having trust, valuing research, being committed to self-directed lifelong learning*, and *valuing teamwork*. Additionally, the authors found that knowledge translators were reported throughout the literature as being "*pragmatic and flexible, positive, persuasive, entrepreneurial, proactive, enthusiastic, open-minded, autonomous, independent, self-sufficient and self-motivated, creative, and committed to principles of equity, inclusivity, respect and cultural competence*" (p. 10) [9]. These latter characteristics were labelled *personality traits* and therefore listed separately from attitudes. Yet another label is used by Cranley et al. [25], who identified *attributes* as part of their review on different facilitating roles. These were being *credible, trustworthy, enthusiastic, creative, passionate, empowering, encouraging, approachable, flexible* and *available*, as well as being a *driving force* and *motivating*. A realist review of interventions and strategies to promote

evidence-informed healthcare describes change agents as having a *positive attitude*, and as being *credible*, *organized*, *accountable*, *reflective*, and *accessible* [26]. Fewer but similar characteristics were identified in a review of the literature on technical assistance concluding that technical assistance providers need to be *collaborative*, *strength-based* (i.e., inspiring and reinforcing), and *trustworthy* [127]. A commonality of these findings is that they describe ISPs as needing attitudes supportive of meeting the stakeholders involved in the implementation process within their contexts *and* of actively guiding them towards new territory. While the former may be best met through respect, empathy, pragmatism, approachability, and flexibility, the latter may be best approached based on drive, honesty, independence, persuasion, and proactivity.

Taken together, the literature this review included draws a picture of ISPs as requiring deep specialist knowledge and a mixed set of attitudes promoting their responsiveness to as well as leadership of stakeholders. The increasing number of studies reporting the involvement of ISPs in RSI implementation also reflects that implementation support roles have gained importance in human services–with this review demonstrating that there is merit in integrating the existing knowledge for ISP roles, which may be labelled differently but are otherwise highly similar in purpose and activities. Integrating and using this knowledge further in considering how to best develop, utilize, and evaluate ISPs in health and human services is of concern to both research and practice.

In having integrated multiple bodies of knowledge focused on different ISP roles in this review, we provide researchers engaged in examining the work of ISPs with a knowledge base that can function as a launch pad for more detailed studies of the role of knowledge and attitudes in implementation support work. It lends itself to intentional efforts of knowledge building and attitude development in ISPs and to studies of how such efforts can influence the implementation capacities of service providers in health and human services. It also invites to compare the effectiveness of different ISPs with different profiles of knowledge and attitudes or to examine how the presence of different types of ISP knowledge and attitudes can be secured and brought to bear through implementation support teams rather than individual ISPs. Furthermore, there is a basis for studying the relationship between ISP knowledge and attitudes. The Knowledge, Attitude, and Practice model (KAP) emphasizes the interrelation between knowledge and attitudes and how they, together, impact behavior in practice [128]. It describes a linear relationship, where (a lack of) knowledge affects attitude and attitude in turn influences behavior. The authors argue that knowledge and attitudes are relational and cannot be interpreted independently from each other. Furthermore, they are dynamic concepts and change over time, in that acquisition of knowledge may lead to changed attitudes. How this occurs is highly dependent on the context, including, e.g., resources, the setting, and individuals involved. It was beyond the scope of this review to understand associations and correlations between ISP knowledge and attitudes and the behavioral consequences of such interaction. Questions of interest for future research may therefore be, for example: *Do ISP attitudes change with growing knowledge*? *What are optimal strategies and conditions for ISP knowledge building*? *Can knowledge be actively used to shape attitudes of ISPs—and how*? *Are there specific types of ISP knowledge and attitudes that are particularly important to effectively support implementation and enabling implementation behavior change in others*? Pursuing these and other pertinent questions about ISPs will help to better understand this particular role in evidence implementation and to build effective implementation support capacities for practice.

For those working in implementation practice, i.e., at the frontline of human service delivery or in intermediary organizations [129], it will be clear from this review that ISP roles represent expertise that is not easily built as part of a professional career as, e.g., psychologist, nurse, or social worker. The complex combination of specialist and generalist knowledge and task

oriented as well as interpersonal attitudes draws a picture of ISP competence that goes beyond the basic training and professional development offered through traditional educational institutions operating in the human service sector. Instead, ISPs will often have built their expertise through a mix of formal and informal educational and professional development activities and continuous on-the-job training embedded in different roles and positions that ISPs have held over longer periods of time.

The results from this review can be used as a starting point for considering more systematically how to utilize and/or develop ISP roles. Decision makers in service and intermediary organizations can use the profile of ISP knowledge and attitudes presented here to understand what to look out for when recruiting ISPs or what to consider when training and developing staff with the aim to establish ISP functions in an agency or organization. By recognizing the role of ISPs, its inherent complexity and diverse resource requirements, organizations can prepare for how to best facilitate and support this position in their local context through, e.g., ongoing education and training, supportive infrastructure and networks, or mentor- and leadership. This is even more pertinent in light of the particular characteristics of the ISP role, including to operate at multiple hierarchical levels of an organization or system, to liaise between different individual and organizational stakeholders, and to balance the need to be a team player with the need to always also provide the perspective of an "outsider". These characteristics explain why ISPs need a broad knowledge base and range of attitudes, and simultaneously indicate that the development and use of such knowledge and attitudes requires organizational as well as leadership support.

In planning for such support, it may be of value to keep in mind that the full range of ISP knowledge and attitudes presented here rarely will be found in a single individual. The purpose of this work is not to create unrealistic expectations for and excessive demands on ISPs. Instead, the knowledge and attitude profile may be viewed as containing building blocks for developing implementation support capacities across multiple individuals in an organization, who also may collaborate as a team and thereby complement each other in their expertise and experience. This may also ease the flexible tailoring of ISP offerings to different projects, settings and stakeholder groups, because there is higher probability for a greater bandwidth of knowledge and attitudes across multiple individuals. The findings from this review and related results from the project into which it is embedded [6, 36, 37] provides a knowledge base for setting up implementation support teams representing the relevant complementary knowledge, attitudes, and skills needed to enable and facilitate implementation processes.

Additionally, educational institutions may want to consider this knowledge base for the development of training opportunities targeting ISPs. The lack of educational offerings and opportunities for those operating in implementation practice, rather than science, has been discussed in the literature as a gap that prevents implementation science knowledge from being used in organizations and systems working to establish cultures of EBP [130–133]. Developing such offerings in ways that build on existing specialist education curricula for, e.g., nurses, psychologists, social workers or allied health practitioners and expand this knowledge based on a generalist, multidisciplinary and implementation informed focus can help different health and human service sectors to more easily build the capacities they need in their work to enhance the uptake of evidence in practice.

## Limitations

This article is the result of a systematically conducted integrative review. Multiple limitations need to be considered when interpreting and using study findings.

The integrative review builds on concept saturation, which means that not all available studies reporting on distinct ISP roles (e.g., facilitator) might have been included. The research team recommends readers especially interested in a particular implementation support role to consult the corresponding literature base.

Furthermore, the search string was created using novel and diverse implementation terminology. While we worked to be as inclusive as possible to include a broad range of ISP terms, the fact that the field of implementation science is young and its terminology is not always aligned, studies which used related but different terminology might not have been identified. Readers should also note that this review aggregates findings from studies predominantly conducted in health and mental health, and to a lesser degree in social welfare and education. While some findings may be applicable to multiple sectors, other findings may be sector specific and require adaption when transferred to other settings.

Important to note as well is that the boundaries between knowledge and attitudes are not always clear when presented in different studies. The distinctions made in this article are a result of ongoing discussions in our research team and thus of group interpretation. Additionally, the results of this paper were based on a dichotomy differing between whether particular knowledge and attitudes were present or not in an included study. The quantity or quality of specific types of knowledge or attitude described in studies were not considered. This means that the included visualizations represent the number of articles mentioning a specific type of knowledge or attitude rather than the total number of mentions of knowledge or attitude within and across articles. Although our work reveals minor differences in ISP knowledge and attitudes according to their position, these findings need to be interpreted with caution, as they are based on a small number of studies. Moreover, this review focused solely on attitudes of ISPs defined as *active mindsets* or *perception filters* as opposed to characteristics or attributes in general.

Finally, studies included in this review used a variety of, e.g., observational, quantitative, qualitative and other research methods. These methods influence study results in that, for example, it makes a difference whether attitudes are self- or externally assessed, as self-perception might vary from the perception by others. This review did not take such potential differences into account and treated extracted data from across studies in the same way independently of the data collection method used.

## Conclusion

Throughout the literature, ISPs display the use of a broad knowledge base and of a palette of attitudes in their work–aimed at matching both the diverse settings and situations they work in and the, at times, contradicting demands on their support. The most common knowledge areas among ISPs are knowledge about *the clinical practice*, *implementation / improvement practice*, *the local context*, *supporting change processes*, and *facilitating EBP in general*. The most reported attitude themes that characterize ISPs are *motivated / motivating / encouraging / empowering*, *empathetic / respectful / sensitive*, *collaborative / inclusive*, *authentic*, *creative / flexible / innovative / adaptive*, and *frank / direct / honest*. These findings show that ISPs are required to be both flexible and firm, to provide support as well as leadership, and to be specialists as well as generalists in their work with and bridge-building between varying individual and organizational stakeholders and their needs and interests. This complexity and its inherent tension call for a comprehensive and systematic approach to developing the ISP as a professional role and to embedding it in different health and social welfare organizations and systems. Developing this approach collaboratively across implementation science, practice, and education will be a crucial step towards building enhanced implementation capacities in human services.

## Supporting information

**S1 Appendix. Electronic Results Addendum (ERA).** Implementation Support Skills–Findings from a Systematic Integrative Review.
(PDF)

**S2 Appendix. PRISMA and PRISMA-S checklists.**
(PDF)

**S3 Appendix. Overview of main study characteristics.**
(PDF)

**S4 Appendix. Overview of descriptors identified in the included studies categorized in the five knowledge themes (*clinical practice knowledge, implementation / improvement practice knowledge, knowledge about the local context, knowledge about supporting change processes, knowledge about facilitating evidence-based practice in general*).** The publication IDs indicate the corresponding articles in which the descriptors were identified.
(PDF)

**S5 Appendix. Overview of descriptors identified in the included studies categorized in the seven attitude themes *(professional, motivated / motivating / encouraging / empowering attitude, empathetic / respectful / sensitive attitude, collaborative / inclusive attitude, authentic attitude, creative / flexible / innovative / adaptive attitude, frank / direct / honest attitude).*** The publication IDs indicate the corresponding articles in which the descriptors were identified.
(PDF)

## Author Contributions

**Conceptualization:** Allison Metz, Katie Burke, Bianca Albers.

**Data curation:** Leah Bührmann, Pia Driessen, Leah Bartley, Cecilie Varsi, Bianca Albers.

**Formal analysis:** Leah Bührmann, Pia Driessen, Bianca Albers.

**Methodology:** Allison Metz, Katie Burke, Bianca Albers.

**Project administration:** Bianca Albers.

**Supervision:** Bianca Albers.

**Visualization:** Leah Bührmann, Pia Driessen.

**Writing – original draft:** Leah Bührmann, Pia Driessen, Bianca Albers.

**Writing – review & editing:** Leah Bührmann, Pia Driessen, Allison Metz, Katie Burke, Leah Bartley, Cecilie Varsi, Bianca Albers.

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
