## [Decision Letter · Decision Letter 0]

7 Mar 2022

PONE-D-21-27502Knowledge and attitudes of Implementation Support Practitioners - Findings from a systematic integrative reviewPLOS ONE

Dear Leah Bührmann,

Thank you for submitting your manuscript to PLOS ONE. After careful consideration, we feel that it has merit but does not fully meet PLOS ONE’s publication criteria as it currently stands. Therefore, we invite you to submit a revised version of the manuscript that addresses the points raised during the review process. This is a generally well written paper but the reporting of the conclusion and methodology could be improved. Please submit your revised manuscript by 28 March 2022. If you will need more time than this to complete your revisions, please reply to this message or contact the journal office at plosone@plos.org. Please include the following items when submitting your revised manuscript:A rebuttal letter that responds to each point raised by the academic editor and reviewer(s). You should upload this letter as a separate file labeled 'Response to Reviewers'.A marked-up copy of your manuscript that highlights changes made to the original version. You should upload this as a separate file labeled 'Revised Manuscript with Track Changes'.An unmarked version of your revised paper without tracked changes. You should upload this as a separate file labeled 'Manuscript'.We look forward to receiving your revised manuscript.

Kind regards,

Eleanor Ochodo

Academic Editor

PLOS ONE

2. Please note that in order to use the direct billing option the corresponding author must be affiliated with the chosen institute. Please either amend your manuscript to change the affiliation or corresponding author, or email us at plosone@plos.org with a request to remove this option.

3. We note that this manuscript is a systematic review or meta-analysis; our author guidelines therefore require that you use PRISMA guidance to help improve reporting quality of this type of study. Please upload copies of the completed PRISMA checklist as Supporting Information with a file name “PRISMA checklist”.

Reviewers' comments:

Reviewer's Responses to Questions

**Comments to the Author**

1. Is the manuscript technically sound, and do the data support the conclusions?

Reviewer #1: Yes

Reviewer #2: Partly

2. Has the statistical analysis been performed appropriately and rigorously? 

Reviewer #1: N/A

Reviewer #2: Yes

3. Have the authors made all data underlying the findings in their manuscript fully available?

Reviewer #1: Yes

Reviewer #2: Yes

4. Is the manuscript presented in an intelligible fashion and written in standard English?

Reviewer #1: Yes

Reviewer #2: Yes

5. Review Comments to the Author

Reviewer #1: The authors have used a systematic integrative approach to describe the knowledge and attitude of Implementation Support Practitioners. The work builds on previous works by the authors. The introduction section is well done and clearly highlights the gap in consolidation in knowledge and attitudes aspects in Implementation area. The methods, results and discussion are well described.

The potential drawbacks of the study like possibility of omission of key studies, authors stating number of articles describing knowledge and attitude rather than number of ISPs have all been described very well in the study limitations area

Authors have mentioned the seven attitude thematic area in the abstract, these are not numbered like it appears on page 23 of 55. I advise that these thematic areas are numbered in the abstract

Reviewer #2: The authors describe the knowledge and attitudes of Implementation Support Practitioners (ISPs) through a systematic review. The content of the work is important and generally well presented but the methodology of the review is unclear.

Please find below my comments:

1. I find the conclusion vague and not really useful in answering the aim of the review which was to present the knowledge and attitudes of ISPs. Could the authors please revise the conclusion by possibly highlighting the key knowledge and attitude themes?

2. The methodology section could be improved as follows:

a. Please define what is meant by an integrative review in the methods section. I struggled to understand what is meant by an integrative until I read the limitations section. In addition please clarify why this is not a scoping review?

b. Please report the search section according to PRISMA for literature searches ;PRISMA-S(https://www.equator-network.org/reporting-guidelines/prisma-s/)

c. Sources searched for literature were broad including social media. Which social media platforms were searched?

d. Were there any limitations in the search with regard to publication date, language, type of publications?

e. Please state the inclusion and exclusion criteria for studies included in the review. What type of studies were eligible? Type of populations and geographical settings? Type of Intervention or concept? Type of Outcomes?

f. Knowing the type of eligible studies will help understand why only randomized trials were included for quality assessment.

g. How were the searches, data extraction and quality assessment done? Was an systematic review application such as covidence used to conduct the searches? How many reviewers did the study selection and data extraction? Was study selection done independently?

h. Was the data extraction form piloted and standardized? Which software or online platform was used to conduct data extraction?

i. It would be helpful to mention a priori in the methods section how the quality assessment was done. More details about the tool used for quality assessment, criteria for scoring?

j. Being a review, please also state in the analysis section that results were synthesized narratively/qualitatively. This will prevent further queries about quantitative results or analysis.

3. Results section

a. I dont find Table 1 helpful. A table of main study characteristics if applicable is better. Some information about the studies is more useful than listing the references only. Else that table can be moved to the appendices.

b. Figure 1 flow chart. Please specify in the first text box under search that these were electronic databases.

c. The results of the quality assessment are not clearly presented in the manuscript.

6. PLOS authors have the option to publish the peer review history of their article (what does this mean?). If published, this will include your full peer review and any attached files.

Reviewer #1: **Yes: **Godfrey Mutashambara Rwegerera

Reviewer #2: No

---

## [Author Response · Author response to Decision Letter 0]

28 Mar 2022

Dear Editor and Reviewers,

We appreciate the comments from the reviewers and the opportunity to submit a revised version. We incorporated the reviewers’ suggestions and believe that the comments highly contributed to the quality of this manuscript. Responses to your specific comments are detailed below. 

Reviewer #1: 

The authors have used a systematic integrative approach to describe the knowledge and attitude of Implementation Support Practitioners. The work builds on previous works by the authors. The introduction section is well done and clearly highlights the gap in consolidation in knowledge and attitudes aspects in Implementation area. The methods, results and discussion are well described.

The potential drawbacks of the study like possibility of omission of key studies, authors stating number of articles describing knowledge and attitude rather than number of ISPs have all been described very well in the study limitations area.

Authors have mentioned the seven attitude thematic area in the abstract, these are not numbered like it appears on page 23 of 55. I advise that these thematic areas are numbered in the abstract

Response: Thank you very much for this suggestion. We have included enumeration in the abstract and agree that this enhances readability.

Reviewer #2: 

The authors describe the knowledge and attitudes of Implementation Support Practitioners (ISPs) through a systematic review. The content of the work is important and generally well presented but the methodology of the review is unclear.

Please find below my comments:

1. I find the conclusion vague and not really useful in answering the aim of the review which was to present the knowledge and attitudes of ISPs. Could the authors please revise the conclusion by possibly highlighting the key knowledge and attitude themes?

Response: Thank you for this comment. We added a few sentences to the conclusion to highlight the key knowledge and attitude themes identified in this review (p. 30, l. 655 – 661).

2. The methodology section could be improved as follows:

a. Please define what is meant by an integrative review in the methods section. I struggled to understand what is meant by an integrative until I read the limitations section. In addition please clarify why this is not a scoping review?

b. Please report the search section according to PRISMA for literature searches ;PRISMA-S(https://www.equator-network.org/reporting-guidelines/prisma-s/)

c. Sources searched for literature were broad including social media. Which social media platforms were searched?

d. Were there any limitations in the search with regard to publication date, language, type of publications?

e. Please state the inclusion and exclusion criteria for studies included in the review. What type of studies were eligible? Type of populations and geographical settings? Type of Intervention or concept? Type of Outcomes?

f. Knowing the type of eligible studies will help understand why only randomized trials were included for quality assessment.

g. How were the searches, data extraction and quality assessment done? Was an systematic review application such as covidence used to conduct the searches? How many reviewers did the study selection and data extraction? Was study selection done independently?

h. Was the data extraction form piloted and standardized? Which software or online platform was used to conduct data extraction?

i. It would be helpful to mention a priori in the methods section how the quality assessment was done. More details about the tool used for quality assessment, criteria for scoring?

j. Being a review, please also state in the analysis section that results were synthesized narratively/qualitatively. This will prevent further queries about quantitative results or analysis.

Response: Thank you for your comments regarding the methodology section. Please find below our remarks to the points you raised.

Re a. We included more details about the integrative review as a knowledge synthesis method (p. 8, l. 177-178). Our review provides more than an orientation and mapping of the role and attributes of ISPs in the field but rather integrates diverse data sources on ISP roles to develop a holistic understanding of the concept. That differentiates our systematic integrative review from a scoping review.

Re b. In addition to the general PRISMA checklist we have already used, we filled in the PRISMA-S checklist and attached it in the S2 Appendix.

Re c. We have included the social media channels that we have used for our literature search (p. 9, l. 203).

Re d. We have added a table on inclusion and exclusion criteria that includes the language and type of publication (p. 10-11). No limitations were defined for publication dates.

Re e. We have added a table on inclusion and exclusion criteria (p. 10-11).

Re f. We fully agree (also see e.).

Re g. We used the platform Covidence to screen the literature, this detail was added to the manuscript (p. 9, l. 187). All included articles were uploaded to Dedoose to code the data, as stated in the “Data analysis” subsection (p. 12, l. 222). The subsections “Literature search” and “Data evaluation” provide information on how the search and quality assessment (also see i.) were performed, respectively. We have included a paragraph on the constitution of the research team in the beginning of the methodology section (p. 9, l. 183-187).

Re h. The data extraction form was standardized but not systematically piloted. We have used the platform Covidence to screen the literature and Dedoose to code and analyse the data, as stated in the “Data analysis” subsection (p. 12, l. 222).

Re i. We provided more detailed information on the quality assessment and the framework used for this (Hodder et al., 2014) in the “Data evaluation” subsection (p. 11, l. 211-219).

Re j. We included this detail in the “Data analysis” section (p. 13, l. 243-244).

3. Results section

a. I dont find Table 1 helpful. A table of main study characteristics if applicable is better. Some information about the studies is more useful than listing the references only. Else that table can be moved to the appendices.

b. Figure 1 flow chart. Please specify in the first text box under search that these were electronic databases.

c. The results of the quality assessment are not clearly presented in the manuscript.

Response: Thank you for these helpful comments. We included an overview of the main study characteristics in S3 Appendix. This overview also indicates if a particular article reported on knowledge, or attitudes, or both, which was previously presented in Table 1. Table 1 was therefore removed from the manuscript. Figure 1 was amended as requested. We rephrased the paragraph on the quality assessment and highlighted the reference to S1 ERA, where the results of the quality assessment are presented.

---

## [Editor Report · Decision Letter 1]

31 Mar 2022

PONE-D-21-27502R1Knowledge and attitudes of Implementation Support Practitioners - Findings from a systematic integrative reviewPLOS ONE

Dear Leah Buhrmann,

Thank you for submitting your revised manuscript to PLOS ONE. After careful consideration, we feel that it has merit but does not fully meet PLOS ONE’s publication criteria as it currently stands. Therefore, we invite you to submit a revised version of the manuscript that addresses the points raised by the academic editor. Please address the minor comments detailed in the Additional Editor Comments section below.

Kind regards,

Eleanor Ochodo

Academic Editor

PLOS ONE

Journal Requirements:

Additional Editor Comments: Thank you for satisfactorily addressing most of the previous reviewer comments. Please address the following minor comments: 1. In page 9, the authors state that " The researchers double-screened the titles, abstracts, and full-texts independently....." . However, this statement is only about study selection. Please also specifically state that data extraction/evaluation were done independently. The Prisma checklist in the appendix states that how risk of bias assessment (data evaluation in your case) was done is detailed in page 9. But that is not so.   2. Also please add in the Abstract methods section that quality assessment of included studies was done and with which tool. 

---

## [Author Response · Author response to Decision Letter 1]

1 Apr 2022

Dear Eleanor Ochodo,

We appreciate your comments and the opportunity to submit a revised version. Responses to your specific comments are detailed below. 

Additional Editor Comments:

Thank you for satisfactorily addressing most of the previous reviewer comments. Please address the following minor comments:

1. In page 9, the authors state that " The researchers double-screened the titles, abstracts, and full-texts independently....." . However, this statement is only about study selection. Please also specifically state that data extraction/evaluation were done independently. The Prisma checklist in the appendix states that how risk of bias assessment (data evaluation in your case) was done is detailed in page 9. But that is not so. 

Response: Thank you for this comment. To clarify this section, we have moved the information on the screening procedure (“The researchers double-screened the titles, abstracts, and full-texts independently using the platform Covidence. Conflicts were solved by a third member of the team.”, p. 9, l. 189) to the section ‘Data Evaluation’, p. 11, l. 216/217. We added information on the data extraction, as requested, to this paragraph:

“Data was extracted from the included studies by using a standardized data extraction form consisting of 19 items (i.e., study design and aim, method, geography, sector, setting, sampling strategy, sample size, clinical intervention, ISP information, outputs, and outcomes). Data from each included study was extracted by one research team member. The quality of the data extraction was assured by the lead author.” (p. 11, l. 217 - 222)

The PRISMA Checklist (appendix 2) hast been updated accordingly. 

2. Also please add in the Abstract methods section that quality assessment of included studies was done and with which tool. 

Response: Information regarding the quality assessment has been added as requested to the abstract (p. 2, l. 39 – 42): 

“Article screening was performed independently by two researchers, and data from included studies were extracted by a member of the research team and quality-assured by the lead researcher. The quality of included RCTs was assessed based on a framework by Hodder and colleagues.”

Kind regards,

Leah Bührmann

---

## [Editor Report · Decision Letter 2]

11 Apr 2022

Knowledge and attitudes of Implementation Support Practitioners - Findings from a systematic integrative review

PONE-D-21-27502R2

Dear Leah Bührmann,

We’re pleased to inform you that your manuscript has been judged scientifically suitable for publication and will be formally accepted for publication once it meets all outstanding technical requirements.

Kind regards,

Eleanor Ochodo

Academic Editor

PLOS ONE

---

## [Editor Report · Acceptance letter]

3 May 2022

PONE-D-21-27502R2 

Knowledge and Attitudes of Implementation Support Practitioners - Findings From a Systematic Integrative Review 

Dear Dr. Bührmann:

I'm pleased to inform you that your manuscript has been deemed suitable for publication in PLOS ONE. Congratulations! Your manuscript is now with our production department. 

Kind regards, 

on behalf of

Prof Eleanor Ochodo 

Academic Editor

PLOS ONE